# Measurement of the Spectral Efficiency of a Heterogeneous Network Architecture of the NG-PON Type for a Quasilinear Propagation Regime

**DOI:** 10.3390/e24040481

**Published:** 2022-03-30

**Authors:** Anyi Girón, Eliana Rivera, Gustavo Gómez

**Affiliations:** Department of Telecommunications, Faculty of Engineering in Electronics and Telecommunications, University of Cauca, Popayán 190001, Colombia

**Keywords:** spectral efficiency, information theory, next generation optical network, measurements and correlation, quasi-linear transmission

## Abstract

The objective of future optical fiber networks is to provide an efficient infrastructure capable of supporting an increasing and variable number of data traffic generated by the diversification of applications with different speed requirements that the current legacy Line Speed networks Single Line Rate (SLR), with predefined modulation formats, cannot supply, because they do not offer enough flexibility to meet the requirements of the demands with such a wide range of granularities. Therefore, next-generation optical networks will be highly heterogeneous in nature, incorporating mixed modulation formats and Mixed Line Rates (MLR). In this work, an analysis of the measurement of the spectral efficiency of a heterogeneous network architecture of the next-generation passive optical network (NG-PON) type is reported for a quasilinear propagation regime through the use of the equation adapted from Shannon’s information theory and developed by the group from the GNTT Research of the University of Cauca, where it was found that it is better to transmit channels of 10 Gbps and 40 Gbps with robust modulations in MLR networks to make an improvement in the spectral efficiency of the network, achieving the same amount of information in a smaller bandwidth or more information in the same bandwidth.

## 1. Introduction

The emergence of new telecommunications services, such as: 4 K/8 K ultra-HD video, virtual reality, multimedia, and cloud computing among others, and the constant improvement of existing ones, makes it necessary to implement technologies capable of satisfying the demands of both speed and bandwidth; the means of transmission that offers advantages in terms of data transmission speed is optical fiber (O.F.) which is why these technologies base their architecture on this means of transmission. Passive optical network (PON) has become popular as an O.F. access network solution; achieving in its third generation an architecture capable of offering a total network performance of 40 Gbps, thanks to the next generation passive optical networks standard, next-generation passive optical network 2 (NG-PON2). These 40 Gbps links can be achieved through the combination of heterogeneous systems with low data transmission speeds, 2.5 Gbps and 10 Gbps, which allows for analysis of the behavior of the spectral efficiency in a quasilinear propagation regime in each combination.

A passive optical network is a network system that sends the signal all, or almost all, of the way to the final user; it is characterized by having a point-to-multipoint distribution model, point-to-multipoint communication (P2MP) using passive components [1]. The development of PON networks continued with the emergence of gigabit-capable passive optical network (GPON) under the ITU-T-G.984 recommendation, which specifies symmetric and asymmetric transmission speeds of 2.5 Gbps for this technology. Traffic demands of up to 10 Gbps lead to the evolution from GPON to NG-PON, which is divided into two phases. The first phase refers to a PON system with a capacity of 10 Gbps in at least one direction, while, in the second phase, there is great speeds of up to 40 Gbps downstream and 10 Gbps upstream [2].

This is how fiber optic networks aim to support a wide variety of data traffic generated by the diversification of applications with different speed requirements, needs that current legacy single line speed networks (SLR) with predefined modulation formats cannot supply, because they do not offer enough flexibility to meet the requirements of demands with such a wide range of granularities. Therefore, future optical fiber networks are expected to be heterogeneous. When speaking of heterogeneous optical networks, reference is made to those networks that allow the implementation of different access capacities for different modulation formats [3,4].

## 2. Data and Methods

The great challenge of optical networks is to satisfy traffic demand while maintaining or reducing network costs, so for this, not only is it enough to increase capacity, but it is also necessary to improve the general utilization of the width band. A combination of improved transport capacity through higher spectral efficiency and bit rate will be required along with better network utilization, by integrating electrical sub-channel preparation into the transmission system [5].

According to Shannon, the capacity of a channel communication is the maximum bit rate that can be transmitted without error, taking into account the noise, the available bandwidth and the limit power. This implies using strong error correction codes and that the robustness of the encoding increases as we approach the limit capacity [6]. The capacity is given by the Shannon formula, as observed in the Equation (1).
(1)$$C=BLog_{2}(1+\frac{S}{N}),$$
where:*C:* Represents the maximum capacity of the channel in bps.*B*: Represents the channel bandwidth measured in Hz.*S/N*: Represents the signal to noise ratio [7].

Shannon’s Law establishes that the capacity of a system depends on the channel bandwidth and the signal-to-noise ratio. Therefore, a more efficient system with a worse SNR can offer the same performance as a less efficient one, and vice versa [8]. In the case of a dense wavelength division multiplexing (DWDM) system, the performance can be improved by using a wider optical bandwidth, or by increasing the spectral efficiency. For the first option additional amplifiers are usually required so increasing the spectral efficiency is usually the most economical alternative [6].

Thus, spectral efficiency is defined as the maximum achievable bit rate per unit of bandwidth and which will depend on aspects, such as the type of modulation, the propagation regime, the bandwidth occupied by the channel, and the signal ratio to noise; it is measured in bps/Hz, and the higher its value, the better the use of the frequency band used to transmit the data [9,10]. Equation (2) represents how spectral efficiency is calculated.
(2)$$\eta_{\left[bps/Hz\right]}=C_{\left[bps\right]}/B_{\left[Hz\right]},$$
where:*η*_[*bps/Hz*]_: Represents spectral efficiency.*C*_[*bps*]_: Represents capacity.*B*_[*Hz*]_: Represents the bandwidth.

In the GNTT Research group of the University of Cauca, in 2019, the work entitled “Improvement of Spectral Efficiency in DWDM networks at 40 Gbps through the Advanced Modulation Formats DPSK and DQPSK” was presented, an equation that expresses the spectral efficiency in terms of the capacity that could be provided for the transport of a network. This equation is an adaptation of the theoretical information presented by Shannon, however, it is important to note that it was presented only for links of a homogeneous type, constant behaviors, and a linear propagation regime, therefore, it presents certain limitations, as heterogeneous networks are the future of optical networks. 

This information was taken up again in order to address the measurement of spectral efficiency in heterogeneous quasilinear systems in NG-PON network architectures.

For an optical system based on wavelength division multiplexing parameters, such as the number of co-propagating channels, the power in the input signal, the spacing per channel, and the propagation regime are handled. Thus, Equation (1) can be expressed as an adaptation to new generation networks, as follows [9]:(3)$$C_{1\left[bits/s\right]}=B_{ch1}Log_{2}(1+\frac{P_{sch1}}{N_{0}B}),$$
(4)$$C_{2\left[bits/s\right]}=B_{ch2}Log_{2}(1+\frac{P_{sch2}}{N_{0}B}),$$
where:*P*: Represents the power of each input channel.*N*_0_: Represents the noise of the channel.*B:* Represents the bandwidth

In which the *C*_1_ and *C*_2_ channels for transmissions with simple modulations, with only linear-type effects generated by power penalties, differ in their transmission rate.

As long as the linear and non-linear correlation of the propagating channels affect the medium; and only considering the response of the linear type and defining that the relation *N*_0_ does not depend on the change in the wavelength of the channel and maintaining the same output power for the ε multiplexed channels of type *i* and the φ multiplexed channels of type *j*; the net capacity of the system will be given in terms of Equation (5) [9]:(5)$$C_{tot\left[bits/s\right]}=(\sum_{i=1}^{\varepsilon }B_{chi}+\sum_{j=1}^{\varphi }B_{chj})\;Log_{2}(1+\frac{P_{s-ch}}{N_{0}B}).$$

The limit of spectral efficiency can be calculated, if the bandwidth of the occupied channels is less than the spacing of the adjacent channels, which means it is necessary that the lateral bandwidth of each of the spectra symmetrical propagating in the middle is contained within the spacing implemented in the DWDM system. Thus, the spectral efficiency would be given by Equation (6).
(6)$$\eta_{bps/Hz}=\frac{\left(C_{OCh1\left[\frac{bits}{s}\right]}+C_{OCh2\left[\frac{bits}{s}\right]}\right)}{(\frac{B_{OCh1\left[Hz\right]}}{2}+\frac{B_{OCh2\left[Hz\right]}}{2})+\Delta_{v}\left[Hz\right]}.$$

If it is desired to decrease the spacing of adjacent channels to a limit in which intersymbolic interference is not generated depending on the type of modulation and the propagation regime, the maximum total efficiency of a system that implements multiple channels would be represented by [9]:(7)$$\eta_{max(\frac{bps}{Hz})}=\frac{\sum_{i=1}^{\varepsilon }B_{chi}Log_{2}(1+\frac{P_{s-chi}}{N_{0}B})+\sum_{j=1}^{\varphi }B_{chj}Log_{2}(1+\frac{P_{s-chj}}{N_{0}B})}{(\frac{B_{Chi\left[Hz\right]}+\;B_{Chj\left[Hz\right]\;}}{2})+(N-1)\Delta_{v}},$$

Whose measurement and control variables correspond to:C_N_ (Nominal capacity per wavelength): It is presented as the net binary capacity propagated at the access level.C_Total_ (Total capacity of the co-propagating system): It is presented as the sum of the individual capacities of the propagated channels.C_Pro_ (Information propagation capacity per kilometer): It is presented as the relationship between the amount of information and the propagated distance.η_ded_ (Dedicated spectral efficiency per wavelength): It is presented as a net relationship between the nominal capacity and the occupied bandwidth.(*N* − 1)_∆*v*_ (Optical spacing of adjacent channels).η_dist_ (Theoretical Distributed Spectral Efficiency of the system): It is presented as a ratio of the total capacity generated in the system vs. the optical spacing.η_disa_ (Analytical distributed spectral efficiency throughout the system): it is presented as a relation of Equation (5) between the total capacity vs. the real occupied spectrum and the maximum allowable SNR relation [9].

Where, the total capacity of the system is divided over half the lower bandwidth occupied by the first optical channel of the spectral grid (lower frequency), plus half the upper bandwidth occupied by the last optical channel of the spectral grid (higher frequency), plus the channel spacing represented as $\left(N-1\right)\Delta_{v}$ (*N* frequency shifts) [9].

As can be seen in Figure 1, all of the parameters described in Equation (7) are related, so it is possible to manipulate these factors in order to improve spectral efficiency.

Equation (7) was used to define the behavior of the spectral efficiency in a quasilinear system for a network with a heterogeneous architecture of NG-PON type. Table 1 shows the variables to be used in the proposed model.

Future networks require high speeds, such as 100 Gbps. To transport a large amount of traffic, these networks are affected by degradations that alter the scope of the optical distance, require expensive equipment to offer long distances or, failing that, a greater number of regenerators, which translates into expenses. Demands on the network may not require 100 Gbps capacity everywhere, therefore, a cost-effective network design should take advantage of the possibility of mixed line speeds (MLR), in wavelength division multiplexed networks (WDM), which increase the capacity of a wavelength to 100 Gbps and present a trade-off between capacity and range [11]. Managing a type of transparent optical architecture, where the signal remains in the optical domain all the time; therefore, electronic signal processing is not necessary in intermediate nodes or, on the contrary, they can be of the opaque type, where each node has electronic regeneration [12].

Various studies have been developed to improve the performance of MLR networks, among which the separation values between channels stand out, considering [12]: (i) a uniform 50 GHz fixed channel spacing specified by ITU-T, (ii) various channel spacing values for different line speeds to optimize the use of spectrum fiber, or (iii) defining an optimal spaced channel value, which results in the minimum cost of the network. It has been shown that as the spacing increases, the cost of the network in terms of transponders decreases up to a certain optimal channel spacing [13]; the chromatic dispersion coefficient is considered to be 17 ps/nm x km, and the PMD (polarization mode dispersion) coefficient of 0.06 ps/nm x km.

Based on the previous information and previous studies, the base network model to use is unidirectional point-to-point WDM, which incorporates multiple modulation formats, with a fixed grid between channels and mixed line speeds (MLR). It has a uniform physical layer infrastructure, with different types of transponders, which can carry signals of 2.5 Gbps, 10 Gbps, 20 Gbps, and 40 Gbps, multiplexers (MUX), a single-mode fiber (SMF), a compensation module of chromatic dispersion fiber (DCF), a demultiplexer (DEMUX), and receivers (RX) that sequentially coincide with the corresponding transmitters, to generate and analyze the different case studies with a target distance of 80 km to 120 km, according to the G. 691 of the ITU.

In addition, in order to guarantee a quasi-linear propagation environment, where the effects of linear types predominate, which are characterized by being constant in time and depend on the manufacturing and medium characteristics, the decision is made to work with very low powers close to 0 dBm and not implementing on-line amplification processes to avoid the excitation of their penalties, in addition, a good receiver is used, whose sensitivity allows the received power to be low.

In the linear regime, the spectral efficiency limit is independent of chromatic dispersion, because it is possible, in principle, to fully compensate for dispersion at the receiver [6].

In the non-linear regime, impairments limit transmission distances, such as the Kerr effect, which has the greatest impact on channel capacity, where the added optical signal intensity disturbs the refractive index of the fiber, thus modulating the fiber’s refractive index. signals phase. Some research has argued that the capacity of DWDM systems is fundamentally limited by XPM, since as a signal propagates, chromatic dispersion converts the XPM-induced phase modulation into intensity noise [6]. Nonlinear effects derived from the reduction of the space between channels and a greater number of channels will be one of the major impediments in wavelength division multiplexing [14].

Regarding modulation formats, it has been verified that on–off keying (OOK) presents higher performance at low speeds of 10 Gbps or lower, while at high speeds, such as 20 Gbps, 40 Gbps, or 100 Gbps, formats, such as differential phase shift keying/differential quadrature phase shift keying (DPSK/DQPSK) respond well; return-to-zero (RZ) encoding is recommended as it has a better tolerance to non-linearity than the non-return-to-zero (NRZ) format [15].

Table 1 lists the elements of a WDM network.

Table 2 shows the variables to be used in the proposed model, where the heterogeneous network simulation schemes were defined to implement in the Optsim software.

This article will address a heterogeneous network consisting of 4 channels of 2.5 Gbps + 2 channels of 10 Gbps + 1 of 40 Gbps, which allows a complete analysis of the parameters involved in a good use of the optical spectrum to obtain the most out of bandwidth without affecting network throughput.

## 3. Heterogeneous MLR Network of the NG-PON Type

For the assembly of this heterogeneous network, Figure 2, the first 2.5 Gbps channel was focused on 193.7 THz, and the remaining three successively separated by 25 GHz, a value that was concluded, when conducting tests, to identify the minimum spacing between channels so as not to affect signals. Subsequently, the 10 Gbps channel was located separate from the fourth 2.5 Gbps channel by 25 GHz, and, finally, the 40 Gbps channel with a 100 GHz separation. An ideal precompensation of the accumulated total chromatic dispersion was performed in the system 100% using a Bragg grid compensator and a length of 80 km for the distribution network and 500 m of fiber for the last mile was handled, using a power of 0 dBm for the 2.5 Gbps channels and 40 Gbps, and 5 dBm for the 10 Gbps.

Finally, it will work with a target bit error rate (BER) of 10^−9^ or less, according to the ITU-T recommendations.

Figure 3a,b shows the optical spectra in transmission and reception, respectively, of the first 2.5 Gbps channel, where there is evidence of a decrease in signal intensity, but no interference from other channels; the same happens with the three remaining channels.

The 10 Gbps channel, adjacent to the 2.5 Gbps channel four, is affected by it, as shown in Figure 4a,b, while, the second channel has no effects on its spectrum due to the fact that it has enough spacing so that its counterpart and the 40 Gbps channel have no impact on it.

In the 40 Gbps channel, a small de-formation is observed to the left of the optical spectrum, which corresponds to the interference caused by the adjacent 10 Gbps channel. It should be noted that the spacing between the second 10 Gbps channel and the 40 Gbps channel is not sufficient for the lower speed channel not to affect the optical spectrum of the higher speed channel, but it does not affect data transmission, achieving a BER of 10^−40^ for both channels, Figure 5a,b.

In this network model, it was possible to transmit 70 Gbps in a bandwidth of 271.212 GHz, as shown in Figure 6.

In the optical spectrum of the throughput Figure 7, the seven transmitted channels are observed, managing to differentiate the characteristic of the signal of each of the different modulations and speeds, as well as small additional channels which can coincide with an original channel producing distortion or loss of information, as a product of the four-wave mixing (FWM) effect or four-wave mixing, this means that, at some point, the transmitted signals go into phase coincidence.

The receiving eye diagrams of the seven channels and their BER value that can be seen in the Figure 8a,b, Figure 9a,b, Figure 10a,b and Figure 11a,b, in their region of the one mark (upper part of the diagrams), show little data dispersion as a consequence of the noise present in the signal, In the region of the opening of the eye (center of the diagrams), symmetry is shown in the time crosses, which indicates that in this case the chromatic dispersion is not a degrading factor of the system, indicating that the defined parameters in transmission (such as the power), and the medium, such as the compensation and the distance, are adequate so that the signals do not present degradation and are not considerably affected by the linear and non-linear effects of transmissions through optical fiber. In addition, the use of different modulation formats directly impacts noise tolerance while increasing the number of symbols per time unit, thus, Mach–Zehnder modulators are almost exclusively used in 40/100 Gbps transport systems, due to its good performance to control modulations and the possibility of independently modulating the intensity and phase of the optical field.

The spectral efficiency of the 2.5 Gbps, 10 Gbps and 40 Gbps channels was calculated in Table 3, Table 4 and Table 5. Table 6 shows the spectral efficiency analysis of the hybrid system, corresponding to a throughput of 70 Gbps.
$$\eta_{max(\frac{bps}{Hz})}=\frac{\sum_{i=1-4}^{\varepsilon }2.5\;Gbps+\sum_{j=1-2}^{\varphi }10\;Gbps+\sum_{k=1}^{\varphi }40\;Gbps}{(\frac{5.18}{2}GHz)+(4*25\;GHz)+(50\;GHz)+100\;GHz+\left(\frac{138}{2}GHz\right)}$$
$$\eta_{max(\frac{bps}{Hz})}=\frac{(2.5\;Gbps)*4+(10\;Gbps)*2+(40\;Gbps)*1}{321.59\;GHz}$$
$$\eta_{max(\frac{bps}{Hz})}=\frac{70\;Gbps}{321.59\;GHz}$$
$$\eta_{max}=0.2177\frac{bps}{Hz}$$

The spectral efficiency for this heterogeneous network, where a throughput of 70 Gbps is transmitted, did not present the results that were expected in comparison with other heterogeneous networks previously studied where 2.5 Gbps and 10 Gbps channels were combined, reaching a throughput of 40 Gbps, since the spectrum used is increased in vain, because low-speed channels do not contribute significantly to throughput and spectrum is wasted in the required spacing between them.

Table 7 summarizes the most important characteristics of the network model proposed in this article.

Regarding the article Adaptation of information theory for the linear propagation regime of a next generation DWDM optical network, which adapts Shannon’s information theory and proposes an equation for the measurement of spectral efficiency, it differs with the present work of degree, in which the development of said equation was carried out for homogeneous systems, therefore, it has certain limitations, since the future of optical networks will be heterogeneous. Through the application in heterogeneous networks, it was possible to distinguish parameters that allow for the improvement of spectral efficiency, such as type of modulation, power and channel spacing, and increase in the number of propagated channels for high access capacities and large bandwidths, achieving a characterization in the design of the heterogeneous network scheme and advanced modulation formats, which significantly improve the performance of the network, with respect to optical degradations.

In the article Migration Methodologies from DWDM Networks to New Generation Networks of 40 Gbps and 100 Gbps, various technologies, that allow efficient migration for links of 40 Gbps and 100 Gbps, were evaluated, such as advanced modulation formats, coherent reception technologies, and coding, achieving systematic and scalable solutions, as in the present work, which showed the same improvement behavior with robust modulations. It is considered that the greater the robustness of the system, optical speeds of up to 100 Gbps are reached, by making use of amplifiers, high-quality receivers, etc., which were not addressed in this degree work, but can provide a considerable improvement in efficiency. This migration from a 10 Gbps to 40 Gbps network is also addressed in the article Impact on Information Spectral Density in an NG-PON network architecture, in which the Shannon information theory equation is also modified. In order to be adapted to this type of network, it is found that by increasing the capacity of the channel from 2.5 Gbps to 10 Gbps, the information spectral density (ISD) increased; and by increasing the spacing between channels from 25 GHz to 100 GHz, the ISD decreased, that is, at a higher transmission rate. For example, from 10 Gbps to 40 Gbps, the ISD increases. In this article the difference between the spectral density of information and the spectral efficiency is clarified. The first refers to a communications channel with n wavelengths, while the second has a single wavelength; in the present work the Spectral density is the same distributed spectral efficiency that is presented as a ratio of the total capacity generated in the system vs. the optical spacing, validating and reaching the conclusion that this factor can be considerably improved if the spacing of the optical channels is significantly reduced.

## 4. Discussion

The improvement in spectral efficiency in next-generation MLR-type systems is very important, because, by increasing said efficiency in a channel, it is possible to represent the same amount of information in a lower bandwidth, or more information in the same width of belt, without the need to require expensive equipment to offer long distances or, failing that, a greater number of regenerators, which translates into expenses. For a transmission system, the highest spectral efficiency channel for a given bandwidth and an optical signal-to-noise ratio (OSNR) is governed by the Shannon limit, below which reliable transmission is not possible for any transmission rate, since infinite signal quality cannot be provided with respect to the noise floor [6].

The importance of robust modulation formats, such as DPSK, is highlighted in the face of noise phenomena and attenuation losses, where these degradations are not significant enough to affect the performance of the system, which, when combined with formats such as RZ-OOK, can double the transmission capacity without incurring a power penalty due to the effect of chromatic dispersion and PMD, therefore it is widely used to transmit these high bit rates. Higher order modulation formats and super spectral channels are key to maximizing spectral efficiency in next-generation optical transport networks. A flexible DWDM grid is key to efficiently accommodate media channels that require approximately 50 GHz of contiguous spectrum [16].

If you want to use MLR networks in a system, it is recommended to transmit channels of 10 Gbps and 40 Gbps, due to the fact that they have a greater number of bits transmitted in a certain bandwidth, indicating a good performance of the spectral density of the information, translating into spectrally efficient systems.

In previous works, the analysis of homogeneous networks was carried out where the spectral efficiency is lower than that obtained in the heterogeneous network studied in this article. For example, to transmit a throughput of 40 Gbps using 2.5 Gbps channels, you need at least 380.18 GHz and only have a spectral efficiency of 10.52%. To transmit the same throughput, you need four channels of 10 Gbps, 187.25 GHz of spectrum, and then an efficiency of 21.36% is obtained; and with a single 40 Gbps channel they occupy 138 GHz reaching an efficiency of 28.99%.

## 5. Conclusions

Next-generation networks are desirable when carrying a large amount of traffic, however, they are affected by degradations that alter the scope of the optical distance. In addition to the demands on the network, they may not require a capacity of 100 Gbps everywhere, therefore, a cost-effective network design must take advantage of the possibility of mixed line speeds (MLR) in wavelength division multiplexed (WDM) networks. If you want to use MLR networks in a system, it is recommended to stream 10 Gbps and 40 Gbps channels.

The small degradations observed occur as the distance increases, which is due to linear and nonlinear effects, however, with the compensation of the ideal Bragg grating at 100%, a great improvement in the results is obtained without considerably increasing the costs of the DWDM network, the DC effect is one of the most detrimental to the signal, and by transmitting at low powers, it avoids inducing the appearance of nonlinear phenomena.

The transport network is one of the most affected by traffic growth, so it requires an increase in throughput, for this, higher order modulation formats and super spectral channels are key to maximizing efficiency spectral in next-generation optical transport networks, postponing/minimizing costly additional fiber deployments, and reconfigurable optical multiplexer node upgrades.

## Figures and Tables

**Figure 1 entropy-24-00481-f001:**
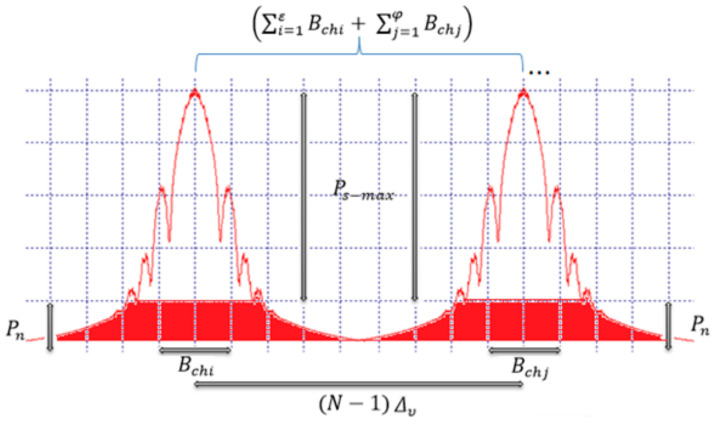
Optical Behavior Relationship in terms of Adapted Information Theory [9].

**Figure 2 entropy-24-00481-f002:**
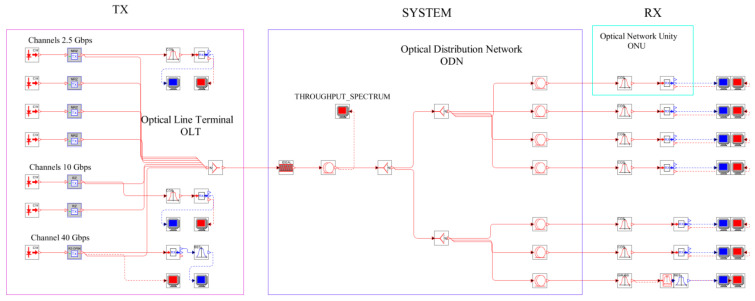
Network scheme.

**Figure 3 entropy-24-00481-f003:**
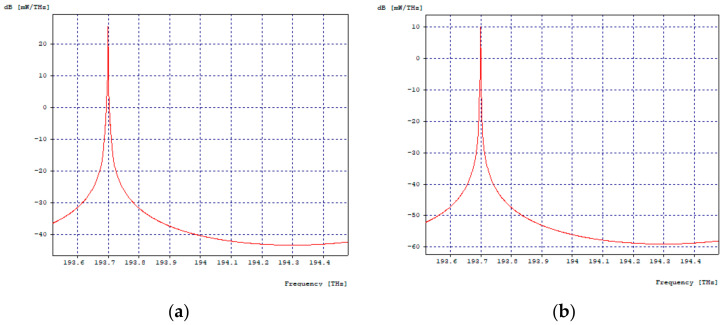
Optical Spectra, First Channel 2.5 Gbps. (**a**) Optical spectrum in transmission; (**b**) Optical spectrum at reception.

**Figure 4 entropy-24-00481-f004:**
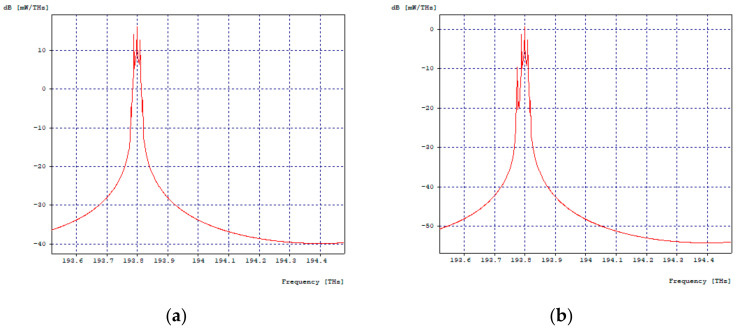
Optical Spectra, 10 Gbps First Channel. (**a**) Optical spectrum in transmission; (**b**) Optical spectrum at reception.

**Figure 5 entropy-24-00481-f005:**
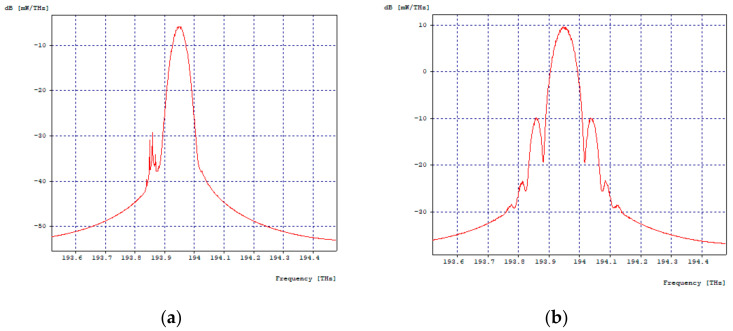
40 Gbps channel optical spectra. (**a**) Optical spectrum in transmission; (**b**) Optical spectrum at reception.

**Figure 6 entropy-24-00481-f006:**
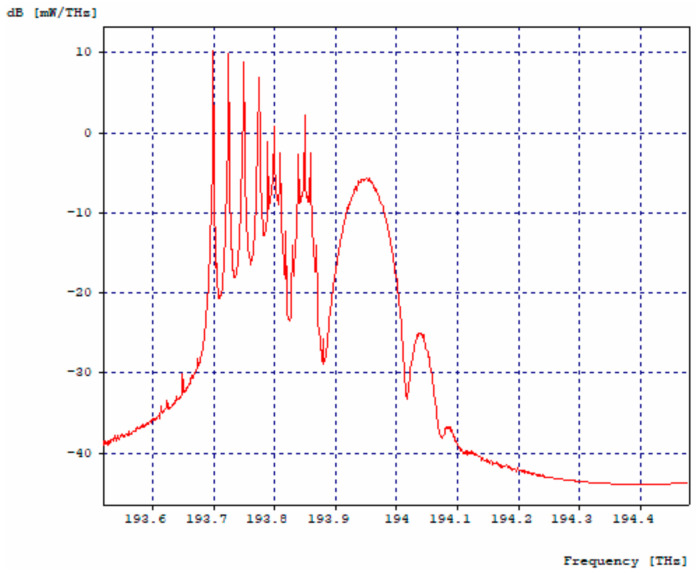
Ideal throughput optical spectrum.

**Figure 7 entropy-24-00481-f007:**
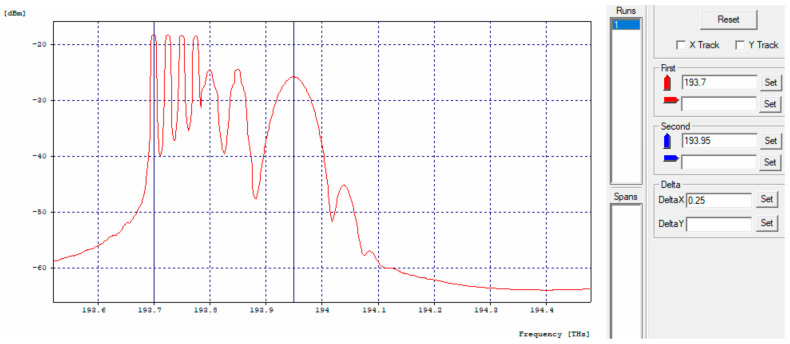
Rectangular throughput optical spectrum.

**Figure 8 entropy-24-00481-f008:**
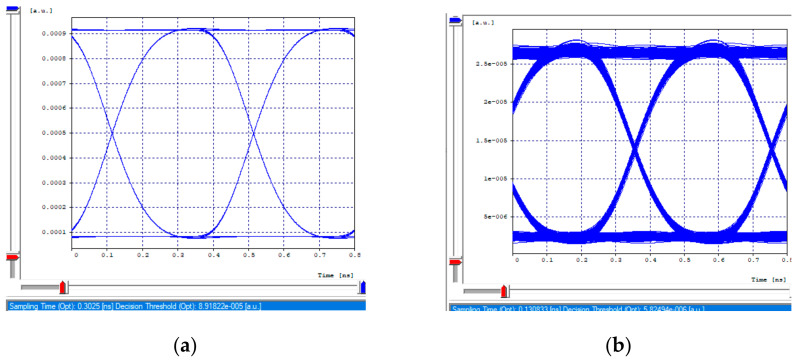
Eye diagram, first channel 2.5 Gbps. (**a**) Transmission eye diagram; (**b**) Eye diagram at reception.

**Figure 9 entropy-24-00481-f009:**
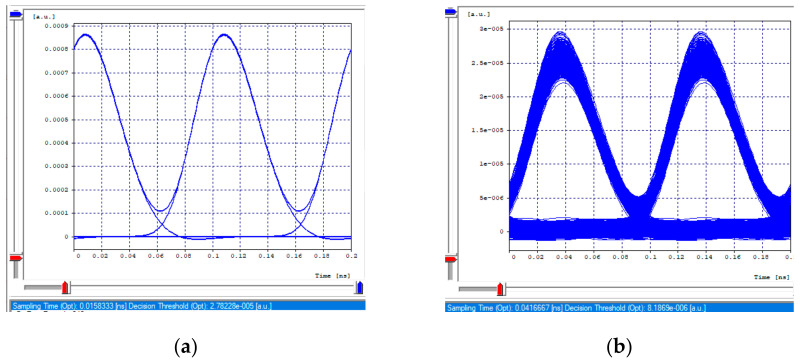
Eye diagrams, first channel 10 Gbps. (**a**) Transmission eye diagram; (**b**) Eye diagram at reception.

**Figure 10 entropy-24-00481-f010:**
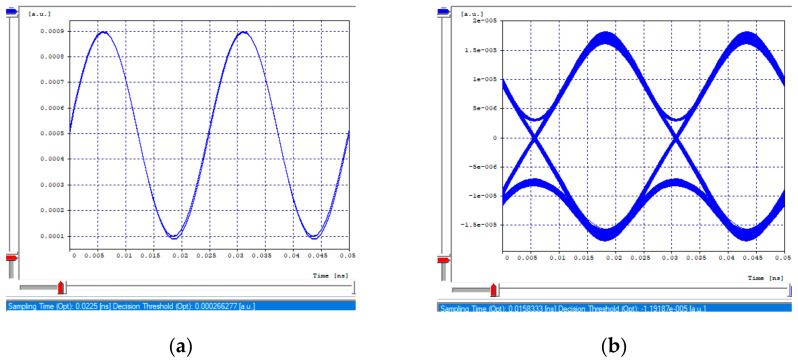
Eye diagrams, 40 Gbps channel. (**a**) Transmission eye diagram; (**b**) Eye diagram at reception.

**Figure 11 entropy-24-00481-f011:**
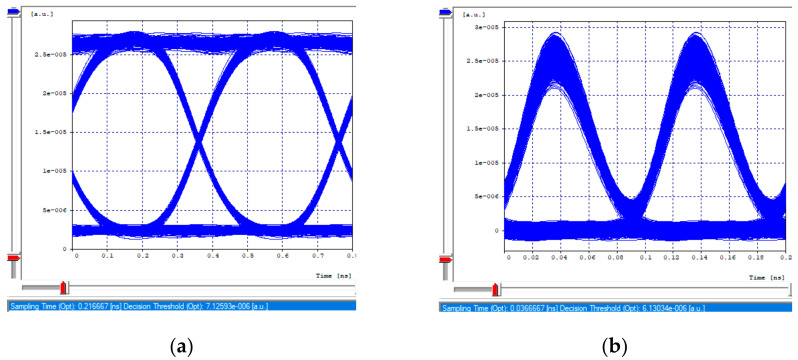
Eye diagrams. (**a**) Eye diagram, channel 3 (2.5 Gbps); (**b**) Eye diagram, channel 2 (10 Gbps).

**Table 1 entropy-24-00481-t001:** Elements of a WDM network.

Section	Elements
TX	- Mach-Zehnder Electric Modulator - Bessel filter - Laser CW 1MHz by FWHM - C band, around 1550 nm
Chanel	- ITU-G652 standard - SSMF-28 fiber - EDFA amplifiers - DCF compensation fiber
RX	- Sensitivity −30dbm - Bessel type electric filter - Elevated cosine optical filter.

**Table 2 entropy-24-00481-t002:** Model variables.

Phases		Model Variables
OLT	Modulation formats	NRZ-OOK, RZ-OOK, RZ-DPSK
	Optical channel power	Maximum allowable to maintain a predominantly quasilinear propagation regime
	Nominal access	2.5 Gbps, 10 Gbps, 40 Gbps
ODN	Optical channel spacing	depends on the factor $\Delta_{v}$
	Link distance	80 km
	Number of channels	Maximum allowable to achieve a higher Throughput of a NG-PON2 network.
	Length extensions	EDFA if required for minimum distance.
	Type of service	FTTH
	Link type	DL- Unidirectional
ONU	Number of users last mile	32
	Splitter	4:1

**Table 3 entropy-24-00481-t003:** Measurement and control variables of the four channels of 2.5 Gbps.

∆v1	C_N_	C_Total_	C_Pro_	η_ded_	η_dist_
25 GHz	2.5 Gbps	4 CH of 2.5 Gbps	Distance 80.5 km	5.18 GHz	80.18 GHz
		10 Gbps	0.81 Tbps/km	9.88%	12.47%

**Table 4 entropy-24-00481-t004:** Measurement and control variables of the two channels of 10 Gbps.

∆v2	C_N_	C_Total_	C_Pro_	η_ded_	η_dist_
50 GHz	10 Gbps	2 CH of 10 Gbps	Distance 80.5 km	37.25 GHz	87.25 GHz
		20 Gbps	1.61 Tbps/km	26.85%	22.92%

**Table 5 entropy-24-00481-t005:** Measurement and control variables, 40 Gbps channel.

∆v3	C_N_	C_Total_	C_Pro_	η_ded_	η_dist_
--	40 Gbps	1 CH of 40 Gbps	Distance 80.5 km	138 GHz	138 GHz
		40 Gbps	3.22 Tbps/km	28.99%	28.99%

**Table 6 entropy-24-00481-t006:** Hybrid system measurement and control variables.

∆v	C_N_	C_Total_	C_Pro_	η_ded_	η_dist_
∆V1 25 GHz	2.5 Gbps	4 CH of 2.5 Gbps	Distance: 80.5 km 5.64 Tbps/km	5.18 GHz	321.59 GHz
		10 Gbps		9.88%	
∆V1 50 GHz	10 Gbps	2 CH of 10 Gbps		37.25 GHz	
		20 Gbps		26.85%	21.77%
∆V3 --	40 Gbps	1 CH of 40 Gbps		138 GHz	
		40 Gbps		232.15%	

**Table 7 entropy-24-00481-t007:** Measurement and control variables of the four channels of 2.5 Gbps.

Characteristics	Chanels 2.5 Gbps				Chanels 10 Gbps	Chanels 40 Gbps
Characteristic eye diagram	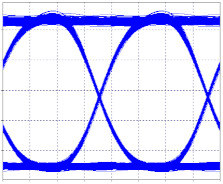				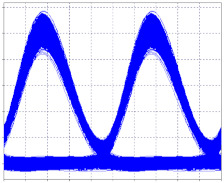	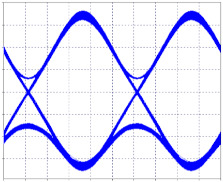
Optical spectrum	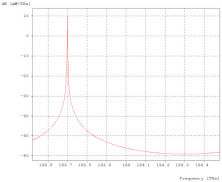				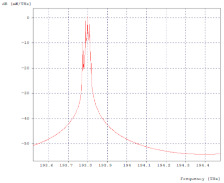	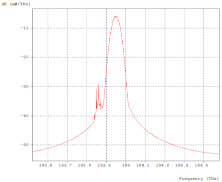
	CH1	CH2	CH3	CH4	CH1	CH1
BER	10^−40^	10^−40^	10^−40^	10^−40^	10^−40^	10^−40^
Optical spectrum throughput	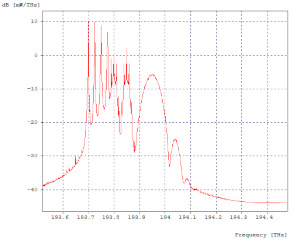					
Spectral efficiency	21.77%					
Tolerance to optical degradation	FWM with low incidence					
Configuration complexity	Complex					
Maximum distance reached	85 km					

## Data Availability

This statement excludes.

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
