# Peer review of "Measurement of the Spectral Efficiency of a Heterogeneous Network Architecture of the NG-PON Type for a Quasilinear Propagation Regime"

_entropy, 2022, doi:10.3390/e24040481_

Round 1

Reviewer 1 Report

The authors present an analysis of the measurement of the spectral efficiency of a heterogeneous network architecture of the NGPON type considering an adaptation of the equation from Shannon's Information Theory and evaluate this proposal using simulation in specific scenarios.

Even though the work presented is interesting and sound, some considerations must be addressed before the article be accepted for publication.

1. The manuscript must be reviewed in its entirety to correct typographical and grammatical errors. Some such errors are listed below.

- The paragraphs in the abstract section must be joined (lines 20 and 21).

- Correct the typo in line 35 - ... bandwidth; The ...

- The paragraphs on lines 45-48 and 49-55 must be together as they represent the same idea.

- Correct the grammar error in line 118.

- Correct the grammar error on line 144.

- Correct the grammar error on line 227.

2. Include a further discussion of Eq. 3, and indicate the difference between Eq. 3 and Eq. 4 because they are the same expression.

3. Include a further discussion of obtaining Eq.5. Include the corresponding demonstration.

4. Edit Fig. 2. This figure is of low quality, and the text on it is unreadable.

5. To simulate a heterogeneous network, the simulator can include different modules that represent sources such as audio and video or other types of signals. These sources can be subjected to various simulation schemes (e.g. QAM).

6. In the simulations, it is possible to use an oscillator to verify the behavior of the signal (in time) before and after transmission. In addition, the discussion part must include a summary table of results indicating the BER obtained and other relevant factors for each scenario.

7. The justification for the simulation scenarios and the selected values is not justified in detail in the manuscript.

8. The manuscript should include a section of related work, and it should indicate the difference and contribution concerning the existing work in the literature.

9. The discussion of results is vague. This section should be expanded and indicate the importance of spectral efficiency in NGPON networks concerning the contribution made.

10. The manuscript must include a section for Conclusions.

Author Response

All recommendations were followed:

1. The manuscript must be reviewed in its entirety to correct typographical and grammatical errors. Some such errors are listed below.we appreciate your time and collaboration (reviewed)

2. Include a further discussion of Eq. 3, and indicate the difference between Eq. 3 and Eq. 4 because they are the same expression. (changes its access and bandwidth, so it corresponds to an MLR system)

3. Include a further discussion of obtaining Eq.5. Include the corresponding demonstration (corresponds to the traffic of a heterogeneous system)

5. To simulate a heterogeneous network, the simulator can include different modules that represent sources such as audio and video or other types of signals. These sources can be subjected to various simulation schemes (e.g. QAM) (It is possible, this network implemented changes in the modulation format and network access)

6. In the simulations, it is possible to use an oscillator to verify the behavior of the signal (in time) before and after transmission. In addition, the discussion part must include a summary table of results indicating the BER obtained and other relevant factors for each scenario (OptSim does not have this type of modules)

7 to 8 (reviewed)

Reviewer 2 Report

The authors investigated the measurement of the spectral efficiency of the next generation optical heterogeneous network architecture for a quasi-linear propagation regime through the adapted equation from Shannon's Information Theory. Results and discussions are interesting and helpful for optical communication networks. This paper can be published in Entropy, provided following issues can be addressed.

  1. Some abbreviations should be clarified when they appear for the first time.
  2. Add a table to specify all parameters used in the transmission network.
  3. For the technical part, the authors employ the Shannon capacity to calculate the spectral efficiency, while actually there some assumptions in the sue of Shannon capacity, e.g. the Gaussian distribution of signals and the Gaussian noise of channel. Please specify the assumptions used in this paper and the validity of them.
  4. Improve the quality of Figure 1 and Figure 2.
  5. In optical communication systems, the chromatic dispersion, the laser phase noise and the fiber nonlinearities will significantly affect the SNR and the spectral efficiency of the optical fiber communication networks. Please add some discussions regarding the impact of dispersion, phase noise, and nonlinear effects on the considered networks to indicate the transmission impairments.

See e.g.

100 Gbps/λ PON downstream O- and C-band alternatives using direct-detection and linear-impairment equalization, Journal of Optical Communications and Networking, 2021.

Achieving high budget classes in the downstream link of 50G-PON, Journal of Optical Communications and Networking, 2021.

Nonlinear coherent optical systems in the presence of equalization enhanced phase noise, Journal of Lightwave Technology, 2021.

Information rates in Kerr nonlinearity limited optical fiber communication systems, Optics Express, 2021.

  1. Discuss the impact of the application of different modulation format on the spectral efficiency in the considered optical heterogeneous network. This is actually also related to the difference between mutual information and Shannon Capacity.

See e.g.

The best modulation format for symmetrical single-wavelength 50-Gb/s PON at O-band: PAM, CAP or DMT?, Optical Fiber Communication Conference, 2021.

High-order modulation formats, constellation design, and digital signal processing for high-speed transmission systems, Optical Fiber Telecommunications VII, 2020.

Author Response

all the recommendations were taken care of, we appreciate your time and collaboration, "Please see the attachment." 

Reviewer 3 Report

Thank you for the submission. This paper presents a theoratical analysis on spectral efficiency of optical transmission system for different data formats. The overall merit is low unfortunately because I couldn't find anything related to state-of-the-art or recent work in this area. Advanced formats should be used in revised edition, theoratical analysis on transmission impairments should be included in estimation of spectral efficiency and a comparision with recently publised papers should be made. If the authors are not planning to propose anything new or different, at least a comparision of different transmission networks or signal processing should be included.

Author Response

(The authors gave the same response as above.)

Round 2

Reviewer 1 Report

The authors of this paper should consider the following comments:

  • Mistake in line 39 the acronym is OF instead of F.O (incorrect order and without a point).
  • Improve the quality in Fig. 2. (unreadable text).
  • The manuscript must include a comparison with other research works in the literature (from other authors) to validate and evaluate the proposed approach. 

Author Response

Corrections made:
• Error on line 39, the acronym is OF instead of FO (wrong order and no period).
• Improve the quality in Fig. 2. (illegible text).
• The manuscript must include a comparison with other research works in the literature (by other authors) to validate and evaluate the proposed approach.

Reviewer 3 Report

Thank you for providing the revisions. I stand by with my previous feedback that the work presented in this paper and the results are very basic and can't be considered to qualify for a research paper. The authors haven't provided a cover letter either but it doesn't affect my decision to not recommend the paper for publication.

Some reviewers have agreed to accept this paper and I leave this to the editor. Thanks a lot

Author Response

(The authors gave the same response as above.)
